

# Testing the mechanism of lepton compositeness

**Vincenzo Afferrante[⋆], Axel Maas[†], René Sondenheimer[‡] and Pascal Törek[○]**

Institute of Physics, NAWI Graz, University of Graz,
Universitätsplatz 5, A-8010 Graz, Austria

⋆ vincenzo.afferrante@uni-graz.at, † axel.maas@uni-graz.at,
‡ rene.sondenheimer@uni-graz.at, ○ pascal.toerek@uni-graz.at

## Abstract

Strict gauge invariance requires that physical left-handed leptons are actually bound states of the elementary left-handed lepton doublet and the Higgs field within the standard model. That they nonetheless behave almost like pure elementary particles is explained by the Fröhlich-Morchio-Strocchi mechanism. Using lattice gauge theory, we test and confirm this mechanism for fermions. Though, due to the current inaccessibility of non-Abelian gauged Weyl fermions on the lattice, a model which contains vectorial leptons but which obeys all other relevant symmetries has been simulated.

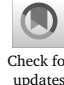
# 1 Introduction

The standard model of particle physics is an exceptionally successful theory [1]. Especially, using elementary particles as physical degrees of freedom in the electroweak sector, motivated by the BRST symmetry in perturbation theory, provides excellent agreement with experiments. However, this is surprising on a deeper field-theoretical level. As these states are gauge-dependent, notwithstanding the Brout-Englert-Higgs (BEH) mechanism, they are strictly speaking unphysical [2,3]. As such, they do not correspond to observable states. This includes even everyday particles like the electron [2,3] and the proton [4].

This troubling paradox can be resolved by the Fröhlich-Morchio-Strocchi (FMS) mechanism [2,3] which is a powerful tool to investigate gauge-invariant information of gauge theories with a BEH mechanism. Physical states of the electroweak sector are necessarily described by gauge-invariant composite objects of the elementary fields, e.g., a scalar particle can be formed by a gauge-invariant bound state of two Higgs doublets. Using conventional gauge-fixing conditions, it can be shown that these bound states behave almost as the elementary particles in the electroweak sector of the standard model. Especially, the FMS mechanism properly maps the pole structure of gauge-dependent states on those of the bound states in a suitable setup. This will be discussed in section 2 in more detail. Furthermore, it can be shown that deviations from the elementary behavior are loop suppressed but affect off-shell properties [5,6]. This mechanism has been confirmed for the $W$, $Z$, and Higgs boson using lattice gauge theory [7,8].

Moreover, it is by now understood that the standard model is special. This can be traced back to the custodial symmetry of the Higgs sector. The FMS mechanism manifests as a one-to-one mapping between gauge multiplets and multiplets of an additional global SU(2) symmetry in the standard model. Qualitative differences occur in general non-Abelian gauge theories with a BEH mechanism already at tree-level in case global symmetries of the Higgs sector and the local gauge group are different [9–11]. Again, this has been quantitatively confirmed for the bosonic sector in lattice simulations [12–14], see Ref. [15] for a detailed review.

While a validation using lattice methods is encouraging, such a radical change of our notion of elementary particles requires an experimental confirmation. Currently, the standard model is the only experimentally tested gauge theory with a BEH mechanism in high-energy physics. Therefore, this task becomes challenging: As noted, differences to usual perturbative treatments are suppressed. Nonetheless, slight deviations exist, are calculable, and have been confirmed again in lattice simulations [5,6,16]. It is therefore, at least in principle, possible to observe them in experiment.

Unfortunately, experiments can neither directly employ electroweak bosons as initial states nor detect them as final states, and thus these slight differences are obscured through production and decay processes. It is thus more natural to investigate the impact on the initial states in collider experiments, i.e., leptons [4,7] and protons [4,17]. The lepton case [2–4,15] will be discussed in more detail in section 2.

While a direct analytical approach may be possible for leptons [4], protons can likely only

be reverse-engineered [17]. In either cases a confirmation of the FMS mechanism in the fermion sector using lattice methods would be a valuable first step, and could provide at least a qualitative insight into expected effects. Unfortunately, this is currently not possible for two reasons. On the one hand, the scales are too separated to be accessible with currently available computer resources. While this could still be controlled using extrapolations, the second problem cannot be circumvented. The lattice regularization introduces a gauge anomaly in the weak interactions, as it is incompatible with parity violation [18]. This problem is unresolved despite many efforts [19–23].

Though a quantitative test is therefore not possible, a qualitative test of the FMS mechanism for fermions is: Parity violation is not an important part of it, and it works in the same way for vectorial fermions. It is thus possible to test the very same mechanism which requires the physical electron to be actually a bound state of an elementary electron and a Higgs field using a vectorial electron. Here, we will perform a first such test.

We do so by investigating a system containing an SU(2) Yang-Mills theory coupled to a gauged scalar doublet mimicking the weak-Higgs subsector of the standard model as well as one generation of vectorial leptons. The latter includes one flavor which is gauged under the weak interaction and two fermion flavors that are ungauged. This allows us to construct a gauge-invariant Yukawa sector which obeys the same pattern as its standard model counterpart. This model, and how the FMS mechanism operates in it, will be introduced in section 3. Its lattice version, the so called Wilson-Yukawa model [24–27] follows in section 4. In this first test, the simulations will be quenched. This is justified as the dynamics of the fermions will not alter the basic principles of the FMS mechanism.

Our primary aims are twofold. On the one hand we want to determine the physical, i. e. gauge-invariant, spectrum of the theory. Thereby, we want to show that there is a genuine composite bound state of the elementary, gauged fermion and the Higgs field. On the other hand we want to demonstrate that the FMS mechanism works and predicts correctly the mass of this state. The corresponding lattice observables will be given in section 5. The results for both items will then be presented in section 6.

We indeed find hints for a positive confirmation of both items and summarize our results in section 7. There, we will also discuss potential next steps as well as implications for experiments.

## 2 Leptons in the standard model

In the following, we rehearse the construction of observables for leptons in the standard model [2–4, 15]. This is particularly useful, as this provides a perspective, especially on the symmetries and degrees of freedom, which is non-standard compared to usual treatments [28]. For simplicity and to be as close as possible to our toy model in section 3, we will treat only the weak interaction, the Higgs doublet, and a single generation of leptons. A generalization to the full standard model can be found in [2, 3, 15]. The other parts of the standard model do not have any relevant influence on the mechanism in the following.

Thus for our purposes, the relevant part of the standard-model Lagrangian is given by

$$
\begin{aligned}
\mathcal{L} = & -\frac{1}{4} W_{\mu\nu}^a W^{a\,\mu\nu} + \frac{1}{2}\mathrm{tr}\left[(D_\mu X)^\dagger (D^\mu X)\right] \\
& -\frac{\lambda}{4}\left(\mathrm{tr}\left(X^\dagger X\right) - v^2\right)^2 + \bar{\psi}^L \mathrm{i}\slashed{D}\psi^L + \bar{\chi}_f^R \mathrm{i}\slashed{\partial}\chi_f^R \\
& -\sum_f y_f \left(\bar{\chi}_f^R\left(X^\dagger\psi^L\right)_f + \left(\bar{\psi}^L X\right)_f \chi_f^R\right).
\end{aligned}
\tag{1}
$$

Herein $W_{\mu\nu}^a$ is the usual field-strength tensor of the weak gauge bosons $W$ and $Z$, and $D_\mu$ is the covariant derivative in the fundamental representation. The matrix-valued field $X$ contains the components of the usual scalar doublet $\phi$ as

$$X = \begin{pmatrix} \phi_2^* & \phi_1 \\ -\phi_1^* & \phi_2 \end{pmatrix}, \tag{2}$$

and thus the standard Higgs fluctuation mode as well as the three would-be Goldstone bosons. The single left-handed Weyl spinor $\psi^L$ is gauged under the weak interaction in the fundamental representation, $\psi^L = (\nu^L \ e^L)^T$. The two flavors of right-handed Weyl spinors $e^R$ and[1] $\nu^R$ are not gauged and combined into a flavor doublet $\chi^R = (\nu^R \ e^R)^T$.

If the Yukawa couplings $y_f$ vanish, the theory obeys three important symmetries.[2] First, we have the local weak gauge symmetry $SU(2)_w$.[3] Second, we have a global $SU(2)_{Rf}$ flavor symmetry of the right-handed Weyl fermions for our particular case. At this point we would like to emphasize that the components of the gauged left-handed spinor cannot be identified with any flavor structure as they merely distinguish different gauge charges similar to the color charge in QCD. Finally, we have a less obvious global $SU(2)_c$ symmetry which acts only on the scalar doublet as a right-multiplication on $X$ and leaves all other fields unchanged. Basically, this symmetry relates the scalar doublet and its charge conjugated counterpart $\epsilon_{ij}\phi_j^*$ (first column of $X$) in a nonlinear way. The advantage of the $X$ notation is the linear realization of $SU(2)_c$ within the standard-model Higgs sector.

If degenerate Yukawa couplings are switched on (within one generation), the global $SU(2)_c$ symmetry of the scalar field and the $SU(2)_{Rf}$ flavor symmetry of the ungauged fermions are broken to a diagonal flavor subgroup $SU(2)_{df}$, which elements $d$ act as $X \to Xd$ and $\chi^R \to d^\dagger \chi^R$.

Ignoring for a moment the BEH effect, there are four physical fermionic states in the theory which are grouped into two chiral doublets. The first two states are the flavor doublet of right-handed Weyl fermions $\chi^R$. One of them is the right-handed charged lepton, $\chi_2^R = e^R$ and the other the right-handed neutrino $\chi_1^R = \nu^R$. The other physical doublet is a gauge-invariant, left-handed Weyl bound state, $\Psi^L = X^\dagger \psi^L$, which is a singlet with respect to the non-Abelian gauge group but carries a global $SU(2)_c$ charge. The two components of this doublet will be identified with the left-handed electron and the left-handed neutrino below. In case non-zero Yukawa couplings break the global $SU(2)_c$ symmetry and the flavor symmetry $SU(2)_{Rf}$ to the diagonal subgroup $SU(2)_{df}$, this bound state transforms in the same way as the right-handed fermions. In this way it appears as if in the physical spectrum the diagonal subgroup acts as an effective flavor symmetry for both the left-handed and right-handed sector. Note that the two gauge-dependent components of the elementary left-handed Weyl fermion $\psi^L$ do not transform under $SU(2)_{df}$, but can be transformed into each other via a gauge transformation and can therefore not be associated with physically observable particles.

Likewise, in the bosonic sector the gauge-invariant bound states $\mathrm{tr}X^\dagger X$ and $\mathrm{tr}\tau^a X^\dagger D_\mu X$ form the physical Higgs and the physical $W$ and $Z$ bosons, a singlet scalar and a triplet vector with respect to $SU(2)_c$, see [2, 3, 15] for details of this sector.

With the BEH effect switched on, and fixing to 't Hooft gauge, the Higgs field can be split into its vacuum fluctuations $\eta$ and its vacuum expectation value. Using the gauge freedom, we conventionally chose the vacuum expectation value to be in the real 2 direction. Thus, we have $\langle \phi_i \rangle = \frac{v}{\sqrt{2}}\delta_{i2}$. At tree-level, this yields the customary result that the gauged and ungauged Weyl spinors can be combined into two Dirac spinors, each with a mass given by $m_f = y_f v/\sqrt{2}$, forming the usual leptons [28]. However, these objects are thus gauge-dependent.

---

[1] We do not consider Majorana neutrinos for simplicity.

[2] How electromagnetic interactions and additional generations fit into the picture can be found in Ref. [15].

[3] Gauge transformations act as a multiplication from the left on the field $X$.

Table 1: The physical states, their quantum numbers, and their leading order contribution in the FMS mechanism.

| Name | Spin | SU(2)$_c$ | SU(2)$_{Rf}$ | Operator | LO FMS expansion |
|---|---|---|---|---|---|
| Higgs | 0 | 0 | 0 | $\mathrm{tr}(X^\dagger X)$ | $\mathrm{tr}(\eta)$ |
| $W/Z$ | 1 | 1 | 0 | $\mathrm{tr}(\tau^a X^\dagger D_\mu X)$ | $W_\mu^a$ |
| Left-handed fermions | $\frac{1}{2}$ | $\frac{1}{2}$ | 0 | $\Psi^L = X^\dagger \psi^L$ | $\psi^{\mathrm{L}} = \begin{pmatrix} \nu^{\mathrm{L}} \\ e^{\mathrm{L}} \end{pmatrix}$ |
| Right-handed fermions | $\frac{1}{2}$ | 0 | $\frac{1}{2}$ | $\chi^R$ | $\chi^R = \begin{pmatrix} \nu^{\mathrm{R}} \\ e^{\mathrm{R}} \end{pmatrix}$ |

Nonetheless, any physical state has to be unaltered by gauge-fixing. Thus only the left-handed bound state $\Psi^{\mathrm{L}}$ and the right-handed $\chi^{\mathrm{R}}$ remain in the fermionic sector. The decisive step is now to realize the FMS mechanism [2,3] to make contact with the conventional and successful perturbative treatment: Expand any gauge-invariant composite operator in the Higgs vacuum expectation value of the scalar field. This yields to leading order the results shown in table 1. As an example, consider the physical left-handed fermion [2,3],

$$\Psi^{\mathrm{L}} = X^\dagger \psi^{\mathrm{L}} = \left( \frac{v}{\sqrt{2}} \mathbb{1} + \eta \right) \psi^{\mathrm{L}} = \frac{v}{\sqrt{2}} \begin{pmatrix} \psi_1^{\mathrm{L}} \\ \psi_2^{\mathrm{L}} \end{pmatrix} + \mathcal{O}(\eta), \tag{3}$$

where the matrix-valued $\eta$ contains the usual fluctuation field identified with the elementary Higgs boson and the Goldstone fields in the same manner as $X$ contains $\phi$ in (2). To leading order $\Psi^L$ thus reduces to the elementary left-handed fermions. The other physical states in table 1 follow in the same way. Of course, only the total sum in (3) is gauge-invariant, and the leading order alone is not.

When now forming a propagator it follows

$$\left\langle \Psi_{f_1}(x)\bar{\Psi}_{f_2}(y) \right\rangle = \frac{v^2}{2} \left\langle \psi_{f_1}^{\mathrm{L}}(x)\bar{\psi}_{f_2}^{\mathrm{L}}(y) \right\rangle + \mathcal{O}(\eta). \tag{4}$$

Thus, to all orders in perturbation theory and to leading order in $\eta$ the propagator of the physical, composite fermion state is given by the gauge-dependent elementary ones, i.e., the propagators of the left-handed charged lepton and neutrino. Especially, the poles and thus the masses coincide. This was shown to all orders in perturbation theory for the Higgs bound-state–elementary-state duality and generalizes straightforwardly to all other standard model particles [5]. In this way, the gauge-invariant Dirac spinor $(\Psi_f^{\mathrm{L}} \ \chi_f^{\mathrm{R}})^{\mathrm{T}}$ describes the physical neutrinos ($f = 1$) and charged leptons ($f = 2$) with the same properties as the usual gauge-dependent ones of perturbation theory [2,3,15]. This can be extended to include the hypercharge sector as well as to quarks [4,15].

As long as the correction $\mathcal{O}(\eta)$ is small, this is an excellent approximation. Indeed, lattice results show this to be the case in the bosonic sector [7,8], though the subleading part is not zero [5,16], and could in principle be accessible even in experiments [4,16,17]. That this is the case is a peculiarity of some theories like the standard model [15,29]. In generic theories, qualitative differences may already appear at leading order [10,11], as confirmed by lattice simulations [12–14].

The important bottom line is that the physical left-handed leptons in the standard model are actually bound states of the elementary ones and the Higgs field [2,3]. That they seem to have the properties of the elementary ones is due to the FMS mechanism, as described in equations (3-4). This could have far-reaching consequences for experiments, if confirmed [4].

The aim here is therefore to test the underlying mechanism. As noted in the introduction, this is not (yet) possible for the standard model (1). Thus, the next step is to create a theory which works in the same way regarding the FMS mechanism, but is accessible to lattice simulations.

## 3 Vectorial leptons

### 3.1 The theory

The chiral nature of the weak gauge theory is the main problem to directly test the FMS mechanism for leptons via lattice simulations. In order to circumvent this technical problem, we investigate a toy model that replaces the Weyl fermions by Dirac spinors. At the same time, our model should be as close as possible to the gauged Higgs-Yukawa structure of the standard model, e.g., via imposing similar internal symmetries. Thus, we investigate a standard-model-like theory with vectorial leptons instead of chiral ones. This may be viewed as an extension where a further generation with opposite helicities is effectively added to one of the standard-model generations. See [26, 27] for earlier discussions of this approach.

More precisely, we study a theory that comprises an $SU(2)_w$ gauge theory coupled to a scalar field and a vectorial fermion $\psi$ in the fundamental representation. Further, we have two flavors of vectorial fermions $\chi$ which are singlets with respect to the gauge group. The latter are coupled to the gauged scalar and fermion field via Yukawa interaction terms. This particular setup can be used straightforwardly in lattice simulations. In particular, the gauged fermion $\psi$ mimics the left-handed leptons of the standard model, while the two components of $\chi$ can be associated with the right-handed electron and neutrino. We will therefore refer to them as vectorial leptons in the following, or just leptons in case it is clear whether the vectorial or standard-model leptons are meant.

The Lagrangian of this theory is given by

$$
\begin{aligned}
\mathcal{L} = &-\frac{1}{4} W_{\mu\nu}^a W^{a\mu\nu} + \frac{1}{2} \mathrm{tr}\left[ (D_\mu X)^\dagger (D^\mu X) \right] \\
&- \frac{\lambda}{4} \left( \mathrm{tr}(X^\dagger X) - v^2 \right)^2 \\
&+ \bar{\psi}\left( i\slashed{D} - m_\psi \right)\psi + \sum_f \bar{\chi}_f \left( i\slashed{\partial} - m_{\chi_f} \right)\chi_f \\
&- \sum_f y_f \left( (\bar{\psi}X)_f \chi_f + \bar{\chi}_f (X^\dagger \psi)_f \right).
\end{aligned}
\tag{5}
$$

It is thus structurally similar to the reduced standard model (1), except that now tree-level masses for the fermions are allowed for the different fermion species. This theory obeys the same symmetries which we discussed for the standard model in Sec. 2, i.e., an $SU(2)_w$ gauge symmetry, a global $SU(2)_c$ symmetry of the scalar field if $y_f = 0$, as well as an $SU(2)_f$ flavor symmetry if $m_{\chi_1} = m_{\chi_2}$ and $y_f = 0$. Except for the additional possibility to break the flavor symmetry by setting $m_{\chi_1} \neq m_{\chi_2}$, the explicit symmetry breaking patterns are the same as in the standard model if we allow for nonvanishing Yukawa couplings or a BEH mechanism via gauge fixing. In the limit of $m_\psi = m_{\chi_f} = 0$ an additional discrete chiral symmetry

$$
\psi \to e^{i\frac{\pi}{2}\gamma_5}\psi, \quad \chi_f = e^{i\frac{\pi}{2}\gamma_5}\chi_f, \quad X \to -X,
\tag{6}
$$

emerges, with $\gamma_5 = i\gamma^0\gamma^1\gamma^2\gamma^3$. As the tree-level masses do not interfere with the FMS mechanism and substantially reduce computing efforts in the lattice simulations, we will consider only finite tree-level masses, and thus this symmetry will be explicitly broken although such terms are forbidden within the standard model.

The physical spectrum contains once more the bound state

$$\Psi = X^\dagger \psi \tag{7}$$

rather than the gauge-dependent $\psi$. It is again a doublet under the global SU(2)$_c$, and what has been discussed in the previous section for the standard model on diagonal flavor symmetry for $y_f \neq 0$ applies here as well. Thus, the theory effectively contains the two vectorial neutrino-like operators $\Psi_1$ and $\chi_1$ as well as the two vectorial electron-like operators $\Psi_2$ and $\chi_2$ in the fermionic sector.

## 3.2 Elementary spectrum

### 3.2.1 Tree-level

Switching now the BEH effect on leads to a somewhat different behavior as in the standard model. Rather than to directly obtain two Dirac particles with separate masses, a non-diagonal mass matrix arises for four Dirac fermions due to the aforementioned doubling of degrees of freedom. Introducing a vector $(\psi\ \chi_1\ \chi_2)^{\mathrm{T}}$, the mass matrix reads

$$M = \begin{pmatrix} m_\psi & 0 & \frac{v}{\sqrt{2}}y_1 & 0 \\ 0 & m_\psi & 0 & \frac{v}{\sqrt{2}}y_2 \\ \frac{v}{\sqrt{2}}y_1 & 0 & m_{\chi_1} & 0 \\ 0 & \frac{v}{\sqrt{2}}y_2 & 0 & m_{\chi_2} \end{pmatrix}. \tag{8}$$

Solving the associated eigenvalue problem leads to four different mass values

$$M_f^\pm = \frac{m_{\chi_f} + m_\psi}{2} \pm \frac{1}{2}\sqrt{(m_{\chi_f} - m_\psi)^2 + 2v^2 y_f^2}. \tag{9}$$

With the five available parameters of the fermion sector, it is possible to form all desired mass values. That these masses do not correspond to the flavors of $\chi$ and $\psi$ is due to the mixing of $\psi_1$ and $\chi_1$ as well as $\psi_2$ and $\chi_2$ caused by the BEH mechanism and the Yukawa couplings. As usual, the corresponding mass eigenstates can be obtained by suitable rotations in field space where we have two different mixing angels $\theta_f$ for the two flavors/components of $\chi_f$ and $\psi_f$.

$$\begin{pmatrix} \zeta_f^+ \\ \zeta_f^- \end{pmatrix} = \begin{pmatrix} \cos\theta_f & \sin\theta_f \\ -\sin\theta_f & \cos\theta_f \end{pmatrix} \begin{pmatrix} \psi_f \\ \chi_f \end{pmatrix},$$

$$\frac{1}{\sin(2\theta_f)} = \sqrt{1 + \frac{(m_\psi - m_{\chi_f})^2}{2y_f^2 v^2}}, \tag{10}$$

where $\zeta_f^\pm$ have mass $M_f^\pm$.

For the present purpose, we select a particular case to reduce the dimensionality of the phase diagram. We set $m_{\chi_f} = m_\psi = m$ and $y_f = y$, and thus reduce the parameter space to two. Then the effective flavor symmetry SU(2)$_{\mathrm{df}}$ is unbroken, and the fermion fields arrange in two doublets with masses

$$M^\pm = m \pm \frac{yv}{\sqrt{2}}. \tag{11}$$

In this particular case, the mass eigenstates are obtained from the charge eigenstates via

$$\zeta^\pm = \frac{1}{\sqrt{2}}(\chi \pm \psi), \tag{12}$$

where $\chi = (\chi_1, \chi_2)^{\mathrm{T}}$.

### 3.2.2 Leading-order quantum corrections

However, beyond tree-level, the situation changes slightly. While the relations $m_{\chi_1} = m_{\chi_2} \equiv m_\chi$ and $y_1 = y_2$ are protected by the diagonal flavor symmetry, the relation $m_\psi = m_\chi$ is not. As the gauged and ungauged fermion flavors couple in different ways to the weak gauge bosons, this leads to a splitting of both mass terms once quantum fluctuations are considered. At the one-loop level, we have

$$
\begin{aligned}
m_\psi^{(1)} &= m(1 + c_y y^2 + c_W \alpha_W), \\
m_\chi^{(1)} &= m(1 + c_y y^2),
\end{aligned}
\tag{13}
$$

where $c_y$ and $c_W$ are dimensionless constants resulting from one-loop integrals with an internal fermion line as well as an internal scalar or gauge boson line, respectively. Further, $\alpha_W = \frac{g^2}{4\pi}$ with $g$ the weak gauge coupling.

Including these one-loop corrections for the mass terms, we obtain for the eigenvalues of the fermion mass matrix

$$
\begin{aligned}
M^\pm &= \frac{m_\psi^{(1)} + m_\chi^{(1)}}{2} \pm \frac{1}{2} \sqrt{\left(m_\psi^{(1)} - m_\chi^{(1)}\right)^2 + 2y^2 v^2} \\
&= m\left(1 + c_y y^2 + \frac{c_W}{2} \alpha_W\right) \pm \frac{1}{2} \sqrt{c_W^2 \alpha_W^2 m^2 + 2v^2 y^2}.
\end{aligned}
\tag{14}
$$

Here, we have neglected one-loop corrections to the Yukawa coupling. These are stronger suppressed in the weak coupling regime as they are $\sim y\alpha_W$ and $\sim y^3$.

As a consequence, the mixing at leading-order (12) will also change. The mixing angle $\theta$ that translates the doublets in the weak charge eigenstates into the mass eigenstates, $\zeta^+ = \psi \cos\theta + \chi \sin\theta$ and $\zeta^- = \chi \cos\theta - \psi \sin\theta$ reads at one-loop order,

$$
\frac{1}{\sin(2\theta)} = \sqrt{1 + \frac{c_W^2 \alpha_W^2 m^2}{2y^2 v^2}}.
\tag{15}
$$

Thus, the maximal mixing of $\psi$ and $\chi$ in the degenerated case $m_\psi = m_\chi$ will be altered. In particular, $\theta$ becomes small if either $y$ is small or $\alpha_W$ is large causing a larger split $m_\psi - m_\chi$.

## 3.3 FMS prediction

The FMS mechanism can be applied in this theory in the same way as in the standard model discussed in section 2. In the bosonic sector this leads to the same results. It gets more interesting for the hybrid $\Psi$. In analogy to (3), the FMS mechanism yields

$$
\Psi = X^\dagger \psi = \frac{v}{\sqrt{2}} \begin{pmatrix} \psi_1 \\ \psi_2 \end{pmatrix} + \mathcal{O}(\eta).
$$

Therefore, the $\Psi$ bound-state can be mapped on the gauge-dependent elementary fermion $\psi$. This implies that $\Psi$ is not a mass eigenstate of the gauge-invariant spectrum as $\psi$ is a linear superposition of $\zeta^\pm$ and we expect two poles at $M^\pm$ for $\langle\Psi\bar\Psi\rangle$. Of course, gauge-invariant mass eigenstates can be constructed from $\Psi$ and $\chi$ via a suitable rotation in field space as in the elementary case. For this purpose, a full analysis of the correlation matrix is required which has to include cross-correlators of the form $\langle\Psi\bar\chi\rangle$ and $\langle\chi\bar\Psi\rangle$ and their corresponding FMS expansions. Treating all FMS expanded terms perturbatively, the FMS mechanism predicts that the mixing of $\chi$ and $\Psi$ is given by the mixing of $\chi$ and $\psi$. This can be most easily seen by choosing on-shell renormalization conditions for the $n$-point functions containing elementary

fields and composite operator insertions following the lines of Ref [5]. In this case it can be shown that the poles and their residues of the gauge-invariant correlators coincide with their gauge-dependent counter parts. Of course, this is a perturbative statement and nonperturbative bound state effects might alter this behavior. However, we do not expect strong deviations in the weak coupling regime.

## 4 Lattice Wilson-Yukawa setup

### 4.1 Dirac operator

In order to discuss the discretization[4] of the theory which we described in section 3.1 we rewrite the fermionic part of the action as [30] [5]

$$
\begin{pmatrix} \bar{\psi} & \bar{\chi} \end{pmatrix} D \begin{pmatrix} \psi \\ \chi \end{pmatrix} = \begin{pmatrix} \bar{\psi} & \bar{\chi} \end{pmatrix} \begin{pmatrix} D^{\bar{\psi}\psi} & D^{\bar{\psi}\chi} \\ D^{\bar{\chi}\psi} & D^{\bar{\chi}\chi} \end{pmatrix} \begin{pmatrix} \psi \\ \chi \end{pmatrix},
\tag{16}
$$

with

$$
\begin{aligned}
D^{\bar{\psi}\psi}_{ij} &= \left(i\slashed{\partial} - m_{\psi}\right)\delta_{ij} - g\,\gamma^{\mu}A^{a}_{\mu}T^{a}_{ij},\\
D^{\bar{\chi}\chi}_{ff'} &= i\slashed{\partial}\,\delta_{ff'} - m_{\chi_1}\delta_{f1}\delta_{1f'} - m_{\chi_2}\delta_{f2}\delta_{2f'},\\
D^{\bar{\psi}\chi}_{if} &= -y_1 X_{i1}\delta_{1f} - y_2 (X^{\dagger})_{2i}\delta_{2f},\\
D^{\bar{\chi}\psi}_{fj} &= (D^{\bar{\psi}\chi})^{\dagger}_{jf},
\end{aligned}
$$

and thus in a block-diagonal form in which the interaction with the Higgs field through the Yukawa interaction explicitly appears in the off-diagonal parts.

To obtain a lattice version, the standard discretization of the bosonic sector is used [30, 33]. For the fermionic action, we give for future reference the most general version, and only specialize afterwards to our case of degenerate parameters. The first diagonal block has then been implemented as a standard SU(2) Wilson-Dirac operator [34]

$$
D^{\bar{\psi}\psi}(x|y)_{ij} = \mathbb{1}\delta_{ij}\delta_{xy} - \kappa_{\psi}\sum_{\mu=\pm1}^{\pm4}(\mathbb{1}-\gamma_{\mu})U_{\mu}(x)_{ij}\,\delta_{x+\hat{\mu},y},
\tag{17}
$$

where $U_{\mu}(x)$ are the links, which satisfy $U_{-\mu}(x) = U_{\mu}(x-\hat{\mu})^{\dagger}$ and $\hat{\mu}$ are unit vectors in the direction $\mu$. With this notation, we have also the relation between the parameters $\kappa_{\psi} = \frac{1}{2(m_{\psi}+4)}$ for the gauged fermion. For the $\gamma$ matrices we used standard chiral Euclidean ones [34].

The second diagonal block has been implemented as a free Wilson-Dirac operator

$$
D^{\bar{\chi}\chi}(x|y)_{ff'} = \mathbb{1}\delta_{ff'}\delta_{xy} - \left(\kappa_{\chi_1}\delta_{f1}\delta_{1f'} + \kappa_{\chi_2}\delta_{f2}\delta_{2f'}\right)\sum_{\mu=\pm1}^{\pm4}(\mathbb{1}-\gamma_{\mu})\delta_{x+\hat{\mu},y},
\tag{18}
$$

where $\kappa_{\chi_f} = \frac{1}{2(m_{\chi_f}+4)}$ are the two hopping parameters for the ungauged fermions.

The third and the fourth block are the Yukawa couplings between the fermions and the Higgs. Due to the Euclidean spacetime they obtain an sign factor, but remain otherwise close

---

[4]From here on all expressions are in lattice notation.

[5]Note that gauge-Higgs-fermion systems without Yukawa interaction have also been investigated on the lattice [31,32].

Table 2: Parameters of the quenched configurations as well as their physical characterization. Quantities without explicit uncertainties have a statistical error below 1%. The scale was set by fixing the mass of the physical $W/Z$ boson, the custodial vector triplet, to $m_{1_3^-} = 80.375$ GeV. $m_{0_0^+}$ is the mass of the scalar bound state $\mathrm{tr}(X^\dagger X)$ corresponding to he Higgs boson of our toy model. The running coupling, and thus the vacuum expectation value, are in the miniMOM scheme [8,37] evaluated at 200 GeV. The results are from the largest volumes employed here, $24^4$.

| # | $\beta$ | $\kappa$ | $\lambda$ | $a^{-1}$ [GeV] | $m_{0_0^+}$ [GeV] | $\alpha_W(200$ GeV$)$ | $v(200$ GeV$) = \frac{m_{1_3^-}}{\sqrt{\pi\alpha_W}}$ [GeV] |
|---|---|---|---|---|---|---|---|
| 1 | 2.7984 | 0.2954 | 1.328 | 384 | 118(9) | 0.544 | 39 |
| 2 | 2.7984 | 0.2978 | 1.317 | 326 | 129(12) | 0.495 | 64 |
| 3 | 3.9 | 0.2679 | 1 | 509 | 116(19) | 0.140 | 121 |
| 4 | 5.082 | 0.249 | 0.7 | 636 | 123(19) | 0.170 | 110 |
| 5 | 5.082 | 0.2552 | 0.7 | 427 | 131(5) | 0.0794 | 161 |

to the continuum form

$$D_{if'}^{\bar\psi\chi}(x|y) = \delta_{xy}\mathbb{1}\left(Y_1 X_{i1}\delta_{1f} + Y_2 X_{i2}^\dagger\delta_{2f}\right),$$
$$D_{f'i}^{\bar\chi\psi}(x|y) = D_{if'}^{\bar\psi\chi\dagger}(x|y). \tag{19}$$

The combined lattice operator is called Wilson-Yukawa operator [24–27].

In the following, we will set $\kappa_F = \kappa_\psi = \kappa_{\chi_1} = \kappa_{\chi_2}$ and $Y = Y_1 = Y_2$. We will furthermore quench the fermionic sector, as will be discussed in more detail in section 4.3. Note that because of the rescaling of the Higgs field and the fermion fields with their own hopping parameters the lattice Yukawa couplings obey $Y_f = y_f\sqrt{\kappa}\kappa_{\chi_f}$.

## 4.2 Inverter

The inverter used in our work is based on the BiCGstab method as discussed in [35]. In particular, our implementation is based on the application of the full operator on a vector $v(x)_{\alpha,i}$ which acts as a fermionic source. The index $\alpha$ is a Dirac index, while the index $i$ indicates the fermionic species, and runs over both the gauge components of $\psi$ and the flavor components of $\chi$, and thus over the four-dimensional Dirac operator (16). The implementation has been checked by calculating the trace of the full operator on a small volume, in the free case with a static Higgs field, and by comparing it with the result from an algebraic computation software. Parallelization has also been enabled for the algorithm, with the openMP API, and tested with respect to the serial results.

For our spectroscopical purposes, we used a single point source located in the origin. This strategy required 16 inversions, given the 4 Dirac indices and the 4 different fermionic species included in the multifield. This point source gave sufficient statistical accuracy for our purposes.

## 4.3 Phase diagram and simulation points

The computational costs for the full theory are, as generically for theories with dynamical fermions, very high. This can already be gathered from simulations without the ungauged fermions in the QCD-like domain [36]. However, as in the standard model, we do not expect a substantial influence of the fermions on the FMS mechanism regarding the mass spectrum of the theory. Thus, we will investigate a quenched scenario in the following for a first qualitative check.

We thus calculate the fermionic spectrum on configurations with dynamical gauge and Higgs fields created using the methods of [8, 33]. This includes a subset of configurations fixed to minimal Landau-'t Hooft gauge, for which the algorithms of [8, 12] were used. These were necessary to determine the propagator of the gauge-dependent field $\psi$. Note that this propagator has much less statistical fluctuations, and we therefore needed only $\mathcal{O}(50)$ configurations for it, while using $\mathcal{O}(1000)$ configurations for the gauge-invariant quantities. As gauge-fixing is very expensive, this leads to roughly the same total computing costs for both types of configurations. However, ultimately the computing time was dominated by the calculation of the fermionic observables below. We list the parameter sets in table 2. They were selected for being suitably similar to the standard-model case in the bosonic sector at either weak coupling or somewhat stronger coupling at different discretizations. However, as will be seen, all parameter sets show essentially the same behavior.

The fermionic sector of the theory described in section 3 has still 5 parameters, three hopping parameters $\kappa_{\psi}$, $\kappa_{\chi_f}$, and the two Yukawa couplings $Y_1$ and $Y_2$. As noted these are reduced to two by $\kappa_F = \kappa_{\psi} = \kappa_{\chi_f}$ and $Y = Y_1 = Y_2$, leaving only $\kappa_F$ and $Y$. It remains to find values for these parameters such that the physics is the one expected for (heavy vectorial) leptons, while at the same time being accessible with available computational resources.

For this purpose we investigated a wide set of $\kappa_F$ and $Y$ values. We found that the time needed for inversion increased substantially for $\kappa_F \gtrsim 1/8$, and thus for negative tree-level masses. At the same time, for $0 < \kappa_F \ll 1/8$ the observed masses in the fermionic sector became very heavy, above one in lattice units. Thus, as a compromise we selected $\kappa_F = 0.11$ and $\kappa_F = 0.12$ as simulation points. For the Yukawa couplings we choose $Y = 0.01$, $Y = 0.05$, and $Y = 0.1$. Again, for larger Yukawa couplings the inversion time became very long. This is to be expected, as the lower mass, according to either (11) or (14), then approaches zero, which yields high inversion times of the Dirac operator. At smaller Yukawa couplings their impact on the masses became too weak to be detectable within our precision. Note that the theory is symmetric under a change of sign of the Yukawa couplings and the Higgs field $X$, and thus we can keep with positive values for the couplings.

To keep finite-volume effects under control, we did simulations for 5 different lattice volumes, $8^4$, $12^4$, $16^4$, $20^4$, and $24^4$. However, we find that even for the finest lattices the infinite-volume behavior has been reached, within available statistical uncertainty, essentially at $20^4$. This is discussed in appendix A in detail, alongside other lattice artifacts. Thus, these choices are sufficient for our purposes. Hence, in total 150 different sets of lattice parameters have been investigated, all in all a little more than 50.000 configurations. The dominant statistical uncertainty stems from the hybrid $\Psi$ bound state, probably due to the strongly fluctuating [8, 33] Higgs field component.

## 5 Spectroscopic observables

Regarding spectroscopy we are interested in the two point functions, which can be accessed from the inverted Wilson-Yukawa operator. We are interested in the bound state $\Psi$, the gauge-invariant fermion $\chi$, and the gauge-dependent fermion $\psi$. Strictly speaking $\Psi$ and $\chi$ have the same quantum numbers, and thus we are looking in principle for the ground state and the first non-trivial excited state. As the excited state could potentially decay, this would require in principle a Lüscher-type analysis. However, as we will see, our results show agreement with the analytical investigation of section 3, implying that the corresponding states are almost stable. This is confirmed in a few cases, where indeed decays and scattering states are kinematically forbidden, and thus we have two stable fermionic states in the physical spectrum. Therefore, a straightforward discretization of the desired propagators of $\psi$, $\chi$, and $\Psi$ turns out to be

sufficient for the purposes at hand.

In the gauge-invariant sector, the full cross-correlation matrix for the two states $\Psi$ and $\chi$ for the propagator is obtained from the Dirac operator (16) and the bound state structure (7) in a straightforward way by Wick contraction [34],

$$M_{\text{GI}}(x|y) = \begin{pmatrix} X^{\dagger}(x)(D^{-1})_{\bar{\psi}\psi}(x|y)X(y) & (D^{-1})_{\bar{\psi}\chi}(x|y)X(y) \\ X^{\dagger}(x)(D^{-1})_{\bar{\chi}\psi}(x|y) & (D^{-1})_{\bar{\chi}\chi}(x|y) \end{pmatrix}. \tag{20}$$

Here we used the fermionic species as subscripts to distinguish the various sections of the inverted operator. They should not be confused with the elements of the Dirac operator in (16). All of the elements of the propagator matrix (20) have a full Dirac matrix structure. For spectroscopy we use the trace in the Dirac structure.

Thus, we get in the off-diagonal blocks the possibility for mixing between the bound states and the elementary fermions, just as with left-handed and right-handed particles in the standard model [4]. Albeit a full variational analysis of (20) reveals that substantially more statistics is required for quantitative precision, we will already get important insights from it. Thus we use the diagonal two propagators individually as well as the eigenvalues from the variational analysis of the full matrix below.

In addition to the gauge-invariant observables we also consider the gauge-dependent $\psi$. In order to obtain a non-zero result for it, it is necessary to invert the operator only on gauge and scalar configurations in a fixed gauge. Conceptually, we have to investigate the cross-correlation matrix for the elementary fermion fields, i.e.,

$$M_{\text{GF}}(x|y) = \begin{pmatrix} (D^{-1})_{\bar{\psi}\psi}(x|y) & (D^{-1})_{\bar{\psi}\chi}(x|y) \\ (D^{-1})_{\bar{\chi}\psi}(x|y) & (D^{-1})_{\bar{\chi}\chi}(x|y) \end{pmatrix}, \tag{21}$$

and compare the results to Eq. (20) to test the FMS mechanism. Of course, matrix (21) is only meaningful within a gauge-fixed setting as otherwise all elements except $(D^{-1})_{\bar{\chi}\chi}$ vanish. Once more, a full variational analysis is expensive and requires more statistics as we have currently available. Thus, we restrict ourselves again on the diagonal elements as they will contain for our purpose all relevant information about the spectrum. As the pure $\chi$-field propagator $\langle \chi \bar{\chi} \rangle$ is gauge-invariant, it does not matter if we calculate it on a gauge-fixed or non-gauge-fixed configuration. Therefore, we use the results of the unfixed configurations for this element and focus in the gauge fixed set up on the Dirac trace of the $(D^{-1})_{\bar{\psi}\psi}$ component.

Note that on our gauge-fixed configurations, where the gauge symmetry and the global symmetry are broken to a common subgroup, it is possible to also define an extended propagator matrix

$$M_{\text{GF}}^{\text{ext}}(x|y) = \begin{pmatrix} X^{\dagger}(x)D^{-1}_{\bar{\psi}\psi}(x|y)X(y) & D^{-1}_{\bar{\psi}\chi}(x|y)X(y) & D^{-1}_{\bar{\psi}\psi}(x|y)X(y) \\ X^{\dagger}(x)D^{-1}_{\bar{\chi}\psi}(x|y) & D^{-1}_{\bar{\chi}\chi}(x|y) & D^{-1}_{\bar{\psi}\chi}(x|y) \\ X^{\dagger}(x)D^{-1}_{\bar{\psi}\psi}(x|y) & D^{-1}_{\bar{\chi}\psi}(x|y) & D^{-1}_{\bar{\psi}\psi}(x|y) \end{pmatrix}, \tag{22}$$

containing the standard propagators of the bound state $\Psi$, the $\chi$ field, and the $\psi$ field on the main diagonal. The off-diagonal elements describe the (gauge-dependent) overlap of these operators. In principle, one could perform a variational analysis on the matrix defined in Eq. (22) to obtain a comprehensive understanding of the fermion sector of this model. However, we are here interested in a first qualitative comparison of the FMS mechanism for fermions. Hence, we will separately analyze (20) and the $(D^{-1})_{\bar{\psi}\psi}$ component of (21) which is sufficient for our task.

As usual, we will perform a zero-momentum projection, which is executed on the inverted matrix elements

$$M(t) = \sum_{\vec{x} \in \Lambda_{\vec{x}}} M(0, \vec{0}|t, \vec{x}). \tag{23}$$

We define the effective masses using the quantity

$$m(t) = -\frac{1}{t - \frac{N_t}{2}} \text{arcosh}\left(\frac{M(t)}{M(N_t/2)}\right). \qquad (24)$$

In this notation, we indicate both the diagonal elements of $M$, so that we have

$$m^{(i)}(t) = -\frac{1}{t - \frac{N_t}{2}} \text{arcosh}\left(\frac{M_{ii}(t)}{M_{ii}(N_t/2)}\right), \qquad (25)$$

but also the masses obtained from the eigenvalues of $M$, which are also called the principal correlators

$$m^{\lambda,i}(t) = -\frac{1}{t - \frac{N_t}{2}} \text{arcosh}\left(\frac{\lambda_i(t)}{\lambda_i(N_t/2)}\right). \qquad (26)$$

The correlators $M(t)$ and $\lambda_i(t)$ show a sum-of-coshs behavior in our finite volumes. As will be seen the correlators contain multiple levels, and thus fits require multiple cosh terms. As a consequence, the effective masses (24) are not perfect straight lines. Furthermore, downward fluctuations in $M(N_t/2)$ yield an upward trend in the effective mass (24). Using the alternative definition

$$m(t) = \ln \frac{M(t)}{M(t+1)}, \qquad (27)$$

which needs to be strictly monotonous decreasing, we confirmed that this is for all physical observables just an artifact of a, probably slightly underestimated, statistical error.

# 6 Spectroscopic results

## 6.1 Impact of quenching on the FMS predictions

For the spectroscopic results, we have in principle clear predictions from the analytical investigations in sections 3.2.2 and 3.3. Nonetheless, a few more comments are in order. On the one hand, lattice simulations will definitely capture more information about the system than the basic one-loop approximations done in Sec. 3. On the other hand, the analytical predictions are derived for fully dynamical fermions while we use a quenched scenario for the simulations in the following to gain a first qualitative investigation of the spectrum. For an actual comparison, we have to either do the FMS analysis within a quenched setting or to perform unquenched simulations. The latter is clearly beyond the scope of this work due to the high computational costs. The former option is challenging as well because a direct translation of the quenched approximation into a continuum formulation is involved.

As a first heuristic step into this direction, we redo our analytic calculation by assuming that the mass of the fermions is much larger than any momentum scale for internal fermion lines. This causes an effective suppression of fermion fluctuations that will properly reproduce the quenching effects in the bosonic sector of the model. However, the situation is less clear for fermionic observables. We find that the mixing effect which exists for dynamical fermions is not captured by the quenched approximation. This manifests in the pole structure of the propagators $\langle \chi \bar{\chi} \rangle$ and $\langle \psi \bar{\psi} \rangle$. For the dynamical case, we have two distinct poles which are present in both propagators. Within our handwaving modeling of the quenched approximation we find that each propagator contains only one pole given by the (quenched) quantum corrected versions of $m_\psi$ and $m_\chi$ for $\langle \psi \bar{\psi} \rangle$ and $\langle \chi \bar{\chi} \rangle$, respectively. In the following, we will denote these infrared mass terms by $M_\psi$ and $M_\chi$ to avoid confusion with the bare mass parameters. We will obtain similar results from the lattice investigations in a moment. Thus, our analysis seems to be consistent from that perspective.

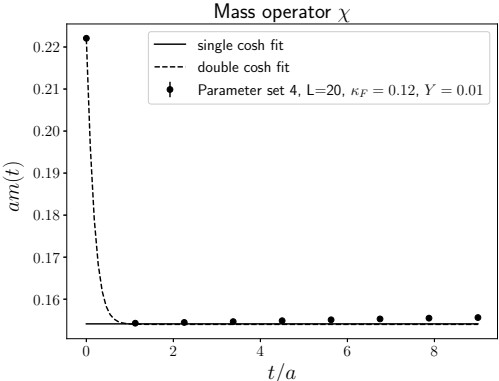 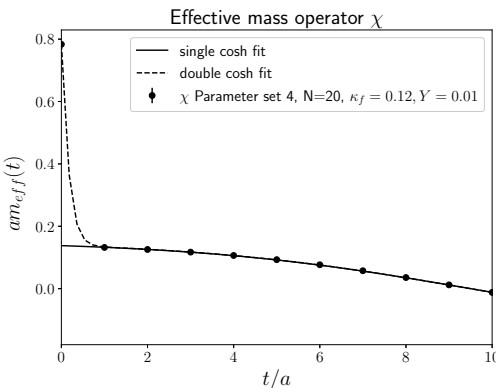

Figure 1: Example of the effective mass from the gauge-invariant propagator of the $\chi$ fermion for system 4 on a $20^4$ lattice at $\kappa_{\mathrm{F}} = 0.12$ and $Y = 0.01$. The fit stems from a two-cosh fit, of which the lighter is also shown alone to emphasize its dominance at long times. Errors are smaller than the symbol size. The top-panel shows the effective mass as defined in (24), while the lower panel shows the effective mass as defined in (27). Note the different scale in both plots.

However, at this point we would like to mention that our modeling of the quenching has some ambiguities when we resum the propagators. For instance, we obtain not only simple pole terms $\sim 1/(p^2 - m^2)$ but also terms of the form $\sim 1/(p^2 - m^2)^2$ as we treat internal and external fermion lines differently. Of course, more sophisticated methods are available to model the quenching, e.g., via introducing additional ghost fields that precisely cancel the fermion determinant which was used in the context of quenched chiral perturbation theory [38–42]. However, the influence of these ghost fields will manifest only within higher loop terms in the fermion propagators when applied to our model and also suffer from unitarity issues.

## 6.2 Lattice results

In the following, we will go carefully through each of our lattice findings. We will exclusively use the infinite-volume extrapolated results, which we obtain along the lines described in appendix A. However, the results on the $20^4$ lattices are already compatible with the infinite-volume results within statistical errors, so this is of little actual concern.

The first interesting question is the gauge-invariant ground state. We find the following pattern. Performing a variational analysis of the effective masses from the gauge-invariant sector, i. e. (20), we find that the lowest level is the same as would be obtained directly from investigating the lower diagonal element, i. e. the correlator of the ungauged fermion $\chi$ alone. This ungauged fermion correlator shows very little noise, and can be very well captured by a double-cosh fit, as shown for an example in figure 1. As discussed in Sec. 5 a very slight upward trend is seen in the effective mass definition (24) not seen in the definition (27), indicating that this is likely a slightly underestimated statistical error.

We therefore conclude that this operator has (essentially) perfect overlap with the ground-state and is the physical lightest particle in this quantum number channel. Because it is a fermionic state, it cannot decay into any of the bosonic particles, and is absolutely stable. Note that in this, and all other channels, the correlator at $t/a \lesssim 2$ is dominated by a mode which has a mass of order $\pi$, and thus is a lattice artifact, which will not be considered further.

The next object is the gauge-fixed propagator of $\psi$, (21). Due to the gauge-dependency [8] we find a (very) slight non-monotonous behavior in the correlator at short times. This is seen

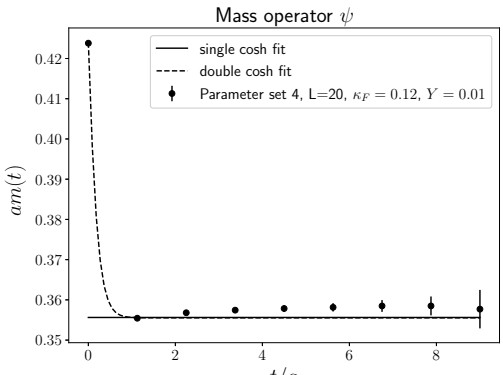
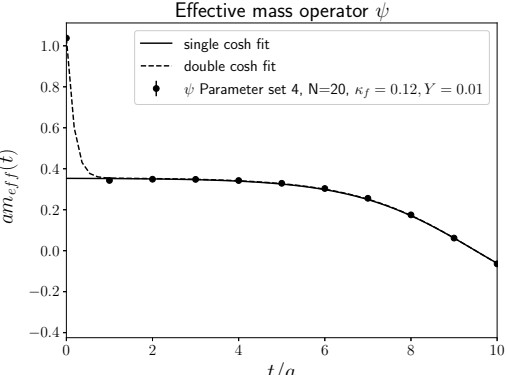

Figure 2: Example of the effective mass from the gauge-fixed propagator (21) for system 4 on a $20^4$ lattice at $\kappa_{\mathrm{F}} = 0.12$ and $Y = 0.01$. The fit stems from a two-cosh fit, of which the lower mass is also shown alone, to emphasis the non-monotonous behavior at short times. Errors are smaller than the symbol size. The top-panel shows the effective mass as defined in (24), while the lower panel shows the effective mass as defined in (27). Note the different scale in both plots.

for the effective mass in both definitions, (24) and (27), and can therefore not be attributed to an underestimation of the correlator at the longest time extent. It signals clearly that this gauge-dependent particle is unphysical. The correlator nonetheless exhibits at late times a good plateau, which is very well described by a single cosh term. This is illustrated in figure 2 for an example. We use this plateau to determine the mass of the gauged fermion. Thus, we conclude that there is only a single state in this channel as well.

The fact that the elementary fields are only dominated by a single mass state is in contradiction to the results of Sec. 3 but as outlined in Sec. 6.1, we trace back this circumstance to the quenching. We find further evidence for this conclusion by the following points. First, we checked the mixed correlators on the diagonal terms of Eq. (21) which describe the overlap of $\psi$ and $\chi$ on gauge-fixed configurations. We indeed find results that are compatible with zero at long times, see Fig. 3.[6] Second, we find that the mass $M_\psi$ extracted from $\langle \psi \bar{\psi} \rangle$ is substantially larger than the mass $M_\chi$. This also fits into the picture as the mass term for $\psi$ gets additional corrections from gauge boson fluctuations which are not present at one-loop order for $M_\chi$, cf. Eq. (13).

The results for the two masses $M_\psi$ and $M_\chi$ are listed in table 3. We find that $M_\chi$ is effectively independent of the gauge coupling and varies only with the Yukawa coupling and indirectly with the parameters of the scalar sector. By contrast, $M_\psi$ depends on the gauge coupling and the ratio $M_\psi/M_\chi$ tends to be smaller with smaller gauge coupling which is expected. The masses are described almost always within $1\sigma$ statistical error by

$$M_\chi = am + r_\chi Y^2, \tag{28}$$

$$M_\psi = am + r_W + r_\psi Y^2, \tag{29}$$

a form motivated by the Taylor expansion at small $y$ of (13). We provide the fit parameters in table 4. Though in principle this can be translated into the form (13), the values should not be identified with the parameters there, as the present ones are the quenched lattice ones.

---

[6]Such a demixing is expected if a random ungauged flavor transformation is applied. Because in the quenched case the two global symmetries are independent at the level of the path integral, and not locked due to the interaction, this is a possible explanation. Conversely, the masses are not affected, and are thus faithfully reproduced.

Table 3: The infinite-volume extrapolated results for the ground-state mass in the gauge-invariant and gauge-dependent channel, which are identified with the masses of the $\psi$ and $\chi$ fermions, see text. The value of $r$ from (30) is given for the $20^4$ lattice, see also appendix A.

| # | $\kappa_F$ | $Y$ | $aM_\chi$ | $aM_\psi$ | $r$ |
|---|---|---|---|---|---|
| 1 | 0.11 | 0.01 | $0.421^{+0.001}_{-0.008}$ | 0.817(3) | 5.7 |
| 1 | 0.11 | 0.05 | 0.407(6) | 0.77(3) | 0.5 |
| 1 | 0.11 | 0.1 | 0.353(9) | 0.54(1) | 0.3 |
| 1 | 0.12 | 0.01 | 0.137(1) | 0.58(1) | 2.1 |
| 1 | 0.12 | 0.05 | 0.111(1) | 0.45(1) | 0.2 |
| 1 | 0.12 | 0.1 | 0.044(5) | 0.21(1) | 0.2 |
| 2 | 0.11 | 0.01 | 0.422(3) | 0.810(4) | 6.1 |
| 2 | 0.11 | 0.05 | 0.406(3) | 0.75(2) | 0.8 |
| 2 | 0.11 | 0.1 | 0.352(2) | 0.62(3) | 0.6 |
| 2 | 0.12 | 0.01 | 0.136(1) | 0.583(4) | 2.6 |
| 2 | 0.12 | 0.05 | 0.103(1) | 0.49(2) | 0.4 |
| 2 | 0.12 | 0.1 | 0.032(2) | 0.17(1) | 0.3 |
| 3 | 0.11 | 0.01 | $0.422^{+0.001}_{-0.006}$ | 0.674(3) | 9.2 |
| 3 | 0.11 | 0.05 | 0.407(5) | 0.645(2) | 0.7 |
| 3 | 0.11 | 0.1 | 0.357(3) | 0.574(4) | 0.2 |
| 3 | 0.12 | 0.01 | 0.136(1) | 0.426(5) | 20.0 |
| 3 | 0.12 | 0.05 | $0.112^{+0.004}_{-0.002}$ | 0.385(2) | 0.4 |
| 3 | 0.12 | 0.1 | 0.043(1) | 0.24(1) | 0.1 |
| 4 | 0.11 | 0.01 | 0.422(1) | 0.604(2) | 17.9 |
| 4 | 0.11 | 0.05 | 0.402(2) | 0.54(1) | 1.2 |
| 4 | 0.11 | 0.10 | 0.331(7) | 0.43(1) | 0.3 |
| 4 | 0.12 | 0.01 | 0.136(3) | 0.346(2) | 767.0 |
| 4 | 0.12 | 0.05 | 0.098(1) | 0.27(2) | 0.8 |
| 4 | 0.12 | 0.10 | 0.036(9) | 0.09(1) | 0.2 |
| 5 | 0.11 | 0.01 | 0.422(5) | 0.599(2) | 10.0 |
| 5 | 0.11 | 0.05 | 0.39(1) | 0.51(1) | 2.0 |
| 5 | 0.11 | 0.1 | 0.305(5) | 0.35(1) | 0.6 |
| 5 | 0.12 | 0.01 | 0.126(4) | 0.347(6) | 315.9 |
| 5 | 0.12 | 0.05 | 0.086(2) | 0.22(1) | 1.3 |
| 5 | 0.12 | 0.1 | 0.03(2) | $0.1^{+0.09}_{-0.05}$ | 0.1 |

Table 4: Fit parameters for $M_{\psi/\chi}$ according to the one-loop motivated fit form (28-29). Note that the fit is done in the lattice Yukawa coupling $Y$ and not the continuum $y$.

| # | $\kappa$ | $am$ | $r_W$ | $r_\chi$ | $r_\psi$ |
|---|---|---|---|---|---|
| 1 | 0.11 | 0.423(8) | 0.41(2) | -6.9(7) | -28.6(2) |
| 1 | 0.12 | 0.1363(5) | 0.43(2) | -9.3(5) | -36.1(1) |
| 2 | 0.11 | 0.423(4) | 0.38(1) | -7.1(2) | -19(3) |
| 2 | 0.12 | 0.1334(9) | 0.46(2) | -10.3(2) | -41.9(1) |
| 3 | 0.11 | 0.423(6) | 0.250(3) | -6.5(3) | -9.9(2) |
| 3 | 0.12 | 0.136(2) | 0.294(4) | -9.3(1) | -18.9(7) |
| 4 | 0.11 | 0.424(1) | 0.172(6) | -9.3(7) | -16.9(7) |
| 4 | 0.12 | 0.131(2) | 0.21(1) | -9.7(8) | -25.4(4) |
| 5 | 0.11 | 0.421(8) | 0.167(6) | -11.7(2) | -24.2(5) |
| 5 | 0.12 | 0.119(2) | 0.197(2) | -9(2) | -22(9) |

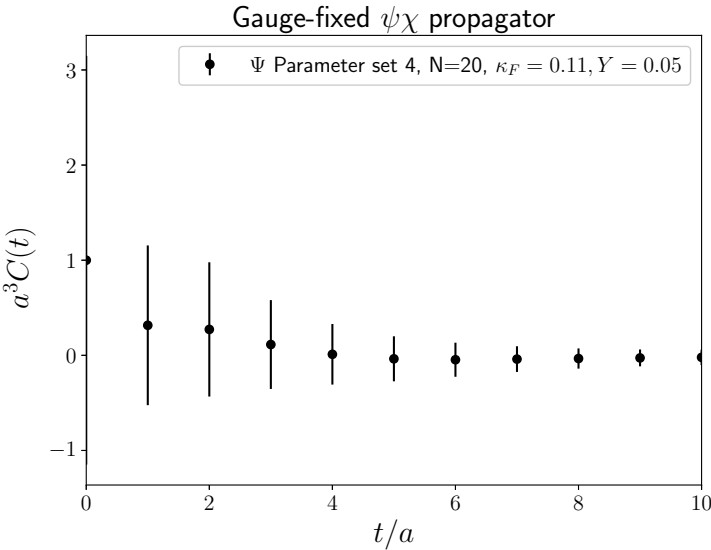

Figure 3: Example for the nonoverlap of $\psi$ and $\chi$ for system 4 on a $20^4$ lattice at $\kappa_F = 0.11$ and $Y = 0.05$.

Further, the fits have been performed in the lattice Yukawa coupling. The actual Yukawa couplings need to be rescaled by $\sqrt{\kappa}\kappa_F \approx 0.06$ on average, giving again reasonable numbers.

Finally, we have to analyze the properties of the bound state operator $\Psi$. This turns out to be quite complicated, especially as it is substantially more noisy than the other two states. However, for the smallest and largest tree-level Yukawa couplings the correlator shows a mass, which is essentially the one of the $\psi$ channel and the $\chi$ channel, respectively. This is also confirmed by the variational analysis of (20), though the errors are considerably larger. This is illustrated in figure 4. We observe again the slightly unphysical behavior of the effective mass of the gauge-dependent $\psi$. Thus the agreement to the physical bound state cannot be exact, but the dominance of the would-be mass is visible in the relevant domain. This subtle comparison of spectral information between physical and unphysical states within the FMS mechanism is discussed in more detail in [5]. We also observe that there is a small admixture of a lighter state at small Yukawa coupling influencing the composite state. Thus, at very late times this level would, of course, dominate.

The origin of this level becomes clear when looking at the intermediate Yukawa couplings, where the situation is different. A naive analysis yields a mass in between the two other channels, with relatively larger errors. However, close scrutiny actually yields that there is a systematic trend of the correlator. The puzzling situation can be resolved by assuming that the bound state correlator is determined by a combination of the two elementary propagators. Thus, a single-parameter fit of type

$$\frac{C(t)}{C(N_t/2)} = \frac{1}{1+r}[\cosh(M_\chi(t - N_t/2)) + r\cosh(M_\psi(t - N_t/2))], \tag{30}$$

with $r$ to be fitted, is performed for late times. This yields a much better agreement. This is also supported by the variational analysis, which indeed finds as second eigenvalue only the mass value $M_\psi$ in the same channel, albeit at substantially larger errors. Reiterating the same procedure also for the other two Yukawa couplings yields that also in that case the correlator is well-described by (30), though extremely strongly dominated by either of the two terms.[7]

---

[7]In addition, we observed non-negligible matrix elements between the different flavor states of $\Psi$ and $\chi$, rather

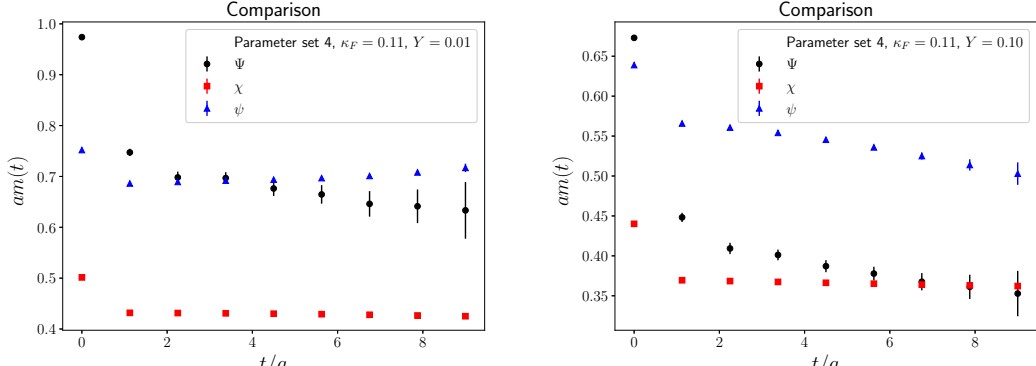

Figure 4: The effective mass of the bound state operator $\Psi$ compared to the effective masses of the elementary fermions $\psi$ and $\chi$ at small Yukawa couplings (top panel) and large Yukawa couplings (bottom panel) for parameter set 4 at $\kappa_F = 0.11$ on $20^4$.

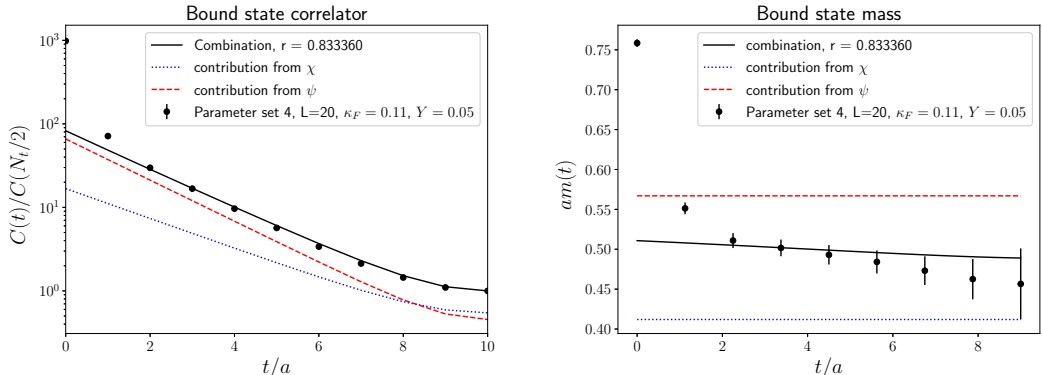

Figure 5: Example of the mixing effect for the bound state correlator (top panel) and the effective mass (bottom panel) for parameter set 4 on $20^4$ at $\kappa_F = 0.11$ and $Y = 0.05$, using the infinite-volume extrapolated masses from table 3.

How this is realized is shown for an example setup in figure 5, and the values for $r$ are also listed in table 3. Note that for many systems the mass difference of both states are less than the mass of the scalar singlet or twice the mass of the vector triplet, and thus the heavier state cannot decay into the ground state, and it is thus stable.

There is a subtlety with determining $r$. The effective mass for the bound state is substantially more noisy than for the elementary fermions. However, (30) limits the upward fluctuations of the effective mass to the error of the larger mass $M_\psi$ and the downward fluctuations likewise to the error of the smaller mass $M_\chi$. Thus, any attempt to find an error band for $r$ would require to allow for relative errors of the input masses in (30) that are larger, and of the same relative size as the one from the bound state correlator itself, and thus than actually are observed and listed in table 2. This would be artificial. The situation is further complicated because of the closeness of the masses $M_{\psi/\chi}$. We therefore quote only an optimal value for $r$ by varying it such that the fit (30) goes best through the effective mass error range of the bound state, as shown in figure 5.

That our results are indeed compatible with the fact that the bound state operator $\Psi$ is a

---

than the same flavor states as obtained at tree-level in (10). This indicates the presence of strong mixing effects due to the Yukawa interactions with the fluctuation mode of the Higgs.

mixture of two mass states that are described by $\psi$ and $\chi$ is consistent with the FMS picture although the quenched analysis makes the interpretation less obvious. As discussed in Sec. 3.3, the FMS mechanism projects the on-shell properties of $\Psi$ onto the the on-shell properties of $\psi$ for a model with dynamical fermions. Thus, $\Psi$ has overlap with two mass eigenstates and the mixing angle of $\Psi$ and $\chi$ is the same as the mixing angle between $\psi$ and $\chi$ at least in a perturbative set up. As we have accumulated evidence that the quenched calculation does not resolve the mixing between the elementary states, one would naively conjecture that the bound state is only described by $M_\psi$. However, a quenched (continuum) FMS analysis would actually be necessary to make a decisive statement. As a quenched continuum analysis is already involved for the elementary fields, a detailed analysis for the FMS mechanism is beyond the scope of this work. Here, we conjecture that the FMS mapping of the mixing of states gets altered as we expect a nontrivial modification of the higher-order FMS terms due to the quenching. In particular the higher-order terms allow for intermediate states that are described by $M_\chi$ and $M_\psi$ causing the observed overlap with both states. The relative ratio between both states is given by the strength of the Yukawa coupling which is confirmed by our data.[8]

We can therefore conclude, that our results support that the physical spectrum is given in terms of the $\chi$ fermion and a bound state of the $\psi$ fermion and the scalar field $X$. The masses of these states are correctly predicted by the FMS mechanism to be the ones of the elementary fermions, both gauge-invariant and gauge-dependent.

Eventually, the physical spectra are shown as a function of the fermionic parameters in figure 6. It is consistent with the FMS prediction in all channels, both bosonic [8], and fermionic, for all parameters. It is also visible that a suitable choice of parameters allows to have both very heavy and very light fermions in the spectrum. In fact, by suitably tuning $\kappa_F$ and $Y$, it appears there would be no obstacle in tuning through the full range of masses from the lightest neutrino to the top quark, and even beyond.

# 7 Conclusions

We have collected evidence that the FMS mechanism is also working in the fermionic sector of gauge-Higgs theories, as anticipated already 40 years ago in [2,3]. Especially, we find that the physical spectrum is indeed a mix of ungauged (would-be right-handed) fermions and (would-be left-handed) fermion-Higgs bound states.

While we have yet investigated a quenched, vectorial system, there is no conceptual difference to the one in the standard model [2–4,15], or even beyond where also gauge-invariant fermion-Higgs bound states are expected [15,43,44]. Furthermore, even though this covered only lepton-like states, the qualitative mechanism is the same also for hadrons [4,15,17]. Of course, this is no guarantee that the mechanism works indeed in all these cases. Ultimate proof will require a detailed analysis of these systems. However, there is no obvious reason that it should not, especially given that the FMS mechanism has passed so far all (lattice) tests [7,8,12–14] in various theories.

This results should serve as a field theoretical foundation for a treatment of fermions in theories with BEH effect, which take fully into account the invariance of the observables under the full gauge group. This includes the standard model. This has far-reaching consequences for phenomenology, as this substructure could be accessible at future colliders [4], like the proposed CLIC [45], FCC-ee [46], or ILC [47]. Identifying and measuring this substructure

---

[8]The lattice also contains nonperturbative information which might not be present within a perturbative treatment of the FMS mechanism. Only an unquenched analysis could reveal as to whether such effects exists and might further modify the mixing.

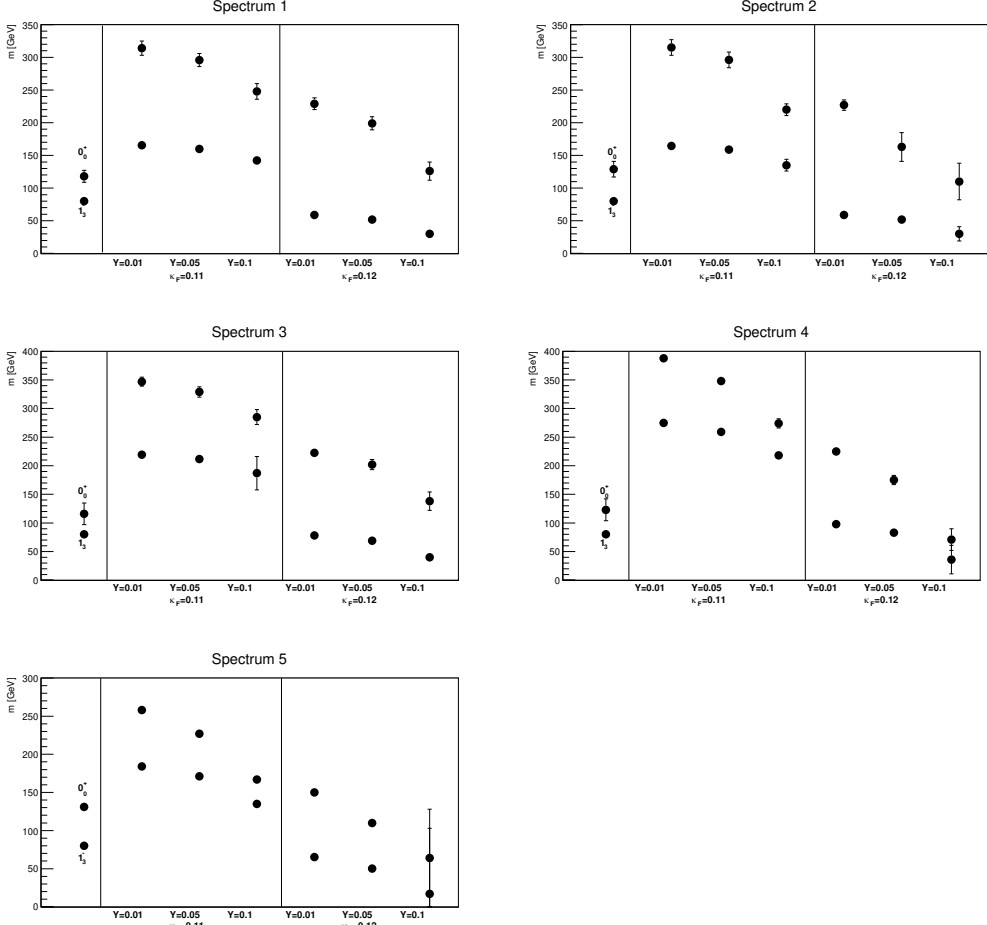

Figure 6: Spectra for all setups. The bosonic sector is fixed, and plotted to the left, while the two states identified in the fermion channel, each being a doublet, are plotted for the different parameters to the right.

easily forms an experimental program in itself. Exploiting the Higgs component also allows to increase the reach for new physics searches which couple to the Higgs component directly, like dark matter through Higgs portals. The results here motivate strongly an experimental program aimed at these effects. Moreover, as it is uniquely tied to the field-theoretical structure of the standard model, it is a guaranteed discovery: Either this effect is found, or there is something very different at work, probably new physics.

A natural next possible step, aside from the obvious but expensive unquenching and reduction of lattice artifacts, is the study of the structure of the Higgs-fermion bound state. Especially form factors, which already illuminated the substructure and size of the vector bosons [16], are an obvious next goal. This should give a first idea of anomalous couplings to the $Z$, as well as the weak radius of the fermion-Higgs bound state[9], both of which are highly interesting questions at future lepton colliders [49].

---

[9]Note that the electromagnetic radius, which appears to be exceedingly small [48], is likely unaffected by the bound state structure, as only the constituent elementary charged leptons carry electromagnetic charge.

## Acknowledgments

V. A. is supported by a FWF doctoral school under grant number W1203-N16 and R. S. by the DFG under grant number SO1777/1-1. The computations have been performed on the HPC clusters at the University of Graz and the Vienna Scientific Cluster (VSC).

## A  Lattice artifacts

There are several possible sources of lattice artifacts in the present calculation. Concerning lattice spacing artifacts, the identification of lines-of-constant physics in the present high-dimensional parameter space is highly non-trivial [33]. However, the main text covers a wide range of lattice spacings for two different cases of weak and strong gauge coupling, without showing any qualitative dependence, and not even a pronounced quantitative one, within statistics. This is consistent with other observables [8, 13, 16, 33], and appears to be rather generic for this type of theory. Though for a detailed quantitative understanding an extended investigation of lines of constant physics will be necessary, at the qualitative level of this work this is sufficient.

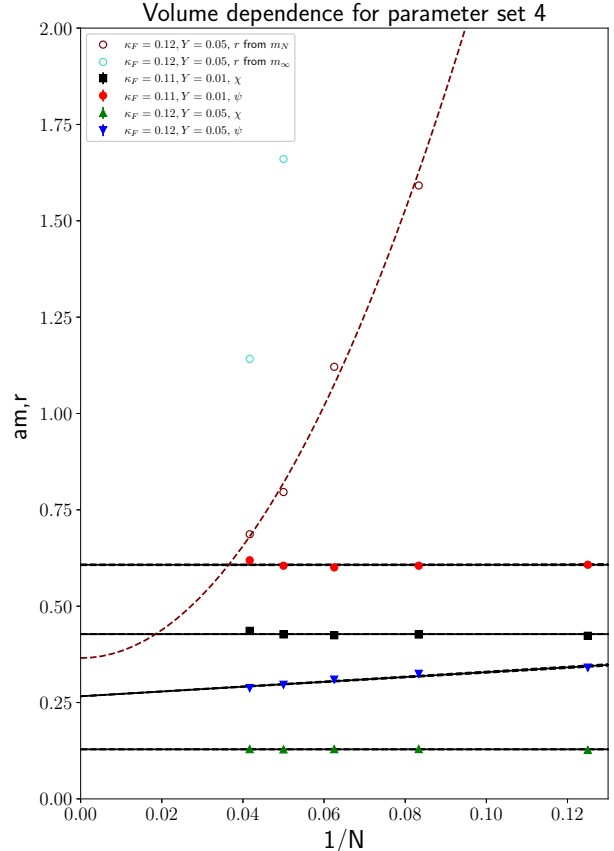

Figure 7: The volume dependence of the masses and $r$ for the parameter set 4 for different values of $\kappa_{\mathrm{F}}$ and two different $Y$ values. The value of $r$ is only shown if substantial mixing occurs. For the masses a fit of type (31) is also shown.

Of course, due to the fact that Wilson fermions require mass renormalization [34], it would be necessary to change the values of the fermion parameters to keep the physical masses fixed when moving along such lines of constant physics. Even in the present case, where the unbroken global symmetry requires some of the parameters to always coincide, this would be a further additional logistical and computational complication. Fortunately, as for the present purpose it is sufficient to have some values of the masses, this is not necessary. When in a next step a continuum extrapolation will be attempted, this will change.

In this context the use of Wilson fermions, which break chiral symmetry explicitly [34], could be problematic. In particular as the mass generation by the BEH effect is, as in the standard model, dynamic. Hence, similar interference as in QCD may be possible [34]. However, in the present model theory we use an additional explicit breaking to avoid negative tree-level masses according to (11). Since the two, quite different, tree-level masses used in the main

text do not show substantial differences in behavior, we conclude that, as in heavy-quark QCD, we are still in a region where we this effect is negligible.

The strongest dependence we see are the dependencies on the volumes. Depending on the lattice parameters, these can be relevant, especially when the comparison of masses of different states is done as needed here. We therefore use a fitting ansatz [50]

$$m_N = m_\infty + \frac{a}{N}e^{-bN}, \tag{31}$$

with the lattice extension $N$ to determine the infinite-volume mass $m_\infty$. This is done using the volumes of $8^4$ to $20^4$, on which the change is largest. The results from the $24^4$ lattices are then used to confirm the fit result. An example for the finest lattice, and thus most extreme volume effects, is shown in figure 7. The results confirm that we see already on the $20^4$ lattice infinite-volume behavior for the masses. In the main text only these infinite-volume masses have been used[10].

The value of $r$ seems to be more dependent on the volume. However, it is visible in figure 7 that the value is substantially dependent on whether the finite-volume masses or the ones extrapolated to infinite volume are used in the fit (30). There is unfortunately no expected behavior for it. However, the results slow down quicker than linear in $1/N$, and a quadratic extrapolation suggests a finite value above zero. Nonetheless, even if the value would be close to zero, or essentially zero, this would only mean that the oscillation effect is strongly volume-dependent, and eventually wins for the intermediate values of the Yukawa couplings.

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
