# Peer review of "Testing the mechanism of lepton compositeness"

_SciPost Physics, doi:SciPost Phys. 10, 062 (2021)_

## Round 2 · Referee Report · Anonymous (Referee 1) · 2020-12-17

• Download as PDF

Report

---

## Round 2 · Referee Report · Anonymous (Referee 2) · 2020-12-25

Report

In this work, the authors try to verify the Frohlich-Morchio-Storocchi mechanism, which shows an equivalence between the gauge singlet bound states and standard particle picture in the gauge fixed perturbation. They simulate gauge-Higgs system with "leptons", which are Dirac fermions, and confirm that FMS mechanism is working. I think this work is interesting enough to be published in SciPost Physics. However, there are some important (most of them are rather technical) issues to be addressed before publication.

  1. It is better to more clearly state what is the FMS mechanism. It is not very clear under what is it and under what condition (always with Higgs?) it is expected to work.

  2. The author states that the standard model is special in that the perturbative and nonperturbative pictures are not very different. They also state that in some theory deviation is seen even at the tree level. I do not understand this at all. More details should be given on, at least, what makes the standard model special.

  3. In table.1, the difference between "tr" and "Tr" is not clear.

  4. In Eq.(6), they show that there exists a discrete chiral symmetry in the chiral limit. But I wonder if this symmetry is anomalous.

  5. 2nd line after Eq.(9): The sentence "That these masses do not..." is difficult to understand.

  6. In Eq.(12), $\chi$ looks undefined.

  7. 2nd line above Eq.(19) what does "overall sign factor" mean?

  8. At the end of 2nd paragraph of page 7, "Note that the theory is symmetric under a change of sign of the Yukawa couplings.." But the sign relative to the Wilson term on a lattice should change the physics.

  9. Eq. (22) What does this extended propagator mean?

  10. Eq (24) and the analysis follows The author should use the (numerical ) solution of cosh(m(t-T/2))/cosh(m(t+1-T/2)) = lattice data, rather than simply taking log.

  11. In page 10 2nd line from the bottom of left column, I cannot understand what "This leaves the bound state Psi" means.

  • validity: -
  • significance: -
  • originality: -
  • clarity: -
  • formatting: -
  • grammar: -

Author:  Vincenzo Afferrante  on 2021-01-12  [id 1140]

(in reply to Report 2 on 2020-12-25)

We thank the referee for her/his careful reading of the manuscript and for her/his comments that have helped improving the presentation of the paper.

In the following, we will address all questions raised by the referee in detail. We also did appropriate changes in the manuscript.

Questions 1. and 2.) *It is better to more clearly state what is the FMS mechanism. It is not very clear under what is it and under what condition (always with Higgs?) it is expected to work.*

*The author states that the standard model is special in that the perturbative and nonperturbative pictures are not very different. They also state that in some theory deviation is seen even at the tree level. I do not understand this at all. More details should be given on, at least, what makes the standard model special.*

In order to address these two questions, we extended the introductory part of the paper to provide the basic idea of the FMS mechanism as well as to emphasize the particular structure of the SM Higgs sector. (See paragraph 2 and 3 in Introduction.)

The referee is right. The FMS mechanism works only for gauge theories with a BEH mechanism. The particularity of the electroweak sector of the standard model is given by the fact that the bosonic sector obeys an additional global SU(2) symmetry besides the local SU(2) gauge group. Within the FMS mechanism the states are classified with respect to the global symmetry group which then can be mapped on the elementary fields (Higgs, W, and Z bosons) which still obey an SU(2) symmetry as the breaking pattern reads $\mathrm{SU}(2) \times \mathrm{SU}(2) \to\mathrm{SU}(2)_{diag}$ (neglecting for simplicity Yukawa interactions and the hypercharge). That all these groups are SU(2) is a particularity of the SM and changes in BSM scenarios.

3.) *In table.1, the difference between "tr" and "Tr" is not clear.*

This is a typo, we changed it consistently to "tr".

4.) *In Eq.(6), they show that there exists a discrete chiral symmetry in the chiral limit. But I wonder if this symmetry is anomalous.*

First of all, we mainly focused on the BEH mechanism and its FMS formulation. Thus this symmetry will be broken spontaneously anyway. Even if a breaking would appear due to quantum corrections, it would be instanton-suppressed in the BEH phase and should have only minor quantitative effects on our results. Furthermore, we break this symmetry explicitly via nonvanishing mass terms for the fermions to minimize computational costs for the inversion of the Dirac operator. We merely highlighted the chiral limit as in this particular case the structure of the Lagrangian of our toy model is the same as the structure of the standard model apart from the fact that we included an additional generation with opposite chirality to obtain vector-like leptons. This particular feature manifests then in this particular discrete global symmetry (note that it also transforms the scalar field, not only the fermions). As the measure of the partition function is invariant under the symmetry, we conclude that this discrete symmetry is not anomalous. Furthermore, there are no AVV triangle diagrams which could spoil the symmetry due to quantum effects (note that these diagrams have to include an internal $\chi$ field line which does not couple to a vector bosons).

In addition, we want to emphasize that our model is free of gauge anomalies as the coupling of the fermions to the gauge bosons is vectorial and there is no Witten anomaly as we have an even number of Weyl fermions.

5.) *2nd line after Eq.(9): The sentence "That these masses do not..." is difficult to understand.*

We wanted to highlight that the fermion flavors in the charge eigenbasis are not mass eigenstates but mix. We modified the paragraph by adding "and reveals a mixing between the flavors $\psi_1$ and $\chi_1$ as well as between the flavors $\psi_2$ and $\chi_2$." below Eq.(8) and canceled the second sentence below Eq.(9).

6.) *In Eq.(12), $\chi$ looks undefined.*

$\chi$ is the flavor doublet of the two uncharged fermion flavors. We dropped the following clause below equation (12) to clarify it: "where $\chi = (\chi_1 \chi_2)^{\mathrm{T}}$".

7.) *2nd line above Eq.(19) what does "overall sign factor" mean?*

By switching from the Minkowski formulation "$- y \bar{\psi}X \chi$" (schematically) of the previous sections to an Euclidean formulation "$+ y \bar{\psi}X \chi$", the momentum-independent interaction part of the Lagrangian gets a global sign. We removed the term "overall" to avoid confusion.

8.) *At the end of 2nd paragraph of page 7, "Note that the theory is symmetric under a change of sign of the Yukawa couplings.." But the sign relative to the Wilson term on a lattice should change the physics.*

We agree that the hopping parameter has to be positive for a well-defined theory. However, our comment should only reflect that $y\to -y$ and $X\to -X$ leaves the Lagrangian in lattice (and continuum) notation unchanged.

9.) *Eq. (22) What does this extended propagator mean?*

The extended propagator describes the the propagation of the $\psi$ field, the $\chi$ field, as well as of the bound state $\Psi$ field. Off-diagonal terms describe potential overlaps of the different operators.

10.) *Eq (24) and the analysis follows. The author should use the (numerical) solution of cosh(m(t-T/2))/cosh(m(t+1-T/2)) = lattice data, rather than simply taking log.*

We actually did use the full lattice data without approximation in the fits as the referee suggests. We merely used the log prescription only for the plots to emphasize the onset of finite volume effects.

11.) *In page 10 2nd line from the bottom of left column, I cannot understand what "This leaves the bound state Psi" means.*

We changed this sentence into: "Finally, we have to analyze the properties of the bound state operator $\Psi$."

---

## Round 2 · Referee Report · Anonymous (Referee 2) · 2020-12-25

Report

In this work, the authors try to verify the Frohlich-Morchio-Storocchi mechanism, which shows an equivalence between the gauge singlet bound states and standard particle picture in the gauge fixed perturbation. They simulate gauge-Higgs system with "leptons", which are Dirac fermions, and confirm that FMS mechanism is working. I think this work is interesting enough to be published in SciPost Physics. However, there are some important (most of them are rather technical) issues to be addressed before publication.

  1. It is better to more clearly state what is the FMS mechanism. It is not very clear under what is it and under what condition (always with Higgs?) it is expected to work.

  2. The author states that the standard model is special in that the perturbative and nonperturbative pictures are not very different. They also state that in some theory deviation is seen even at the tree level. I do not understand this at all. More details should be given on, at least, what makes the standard model special.

  3. In table.1, the difference between "tr" and "Tr" is not clear.

  4. In Eq.(6), they show that there exists a discrete chiral symmetry in the chiral limit. But I wonder if this symmetry is anomalous.

  5. 2nd line after Eq.(9): The sentence "That these masses do not..." is difficult to understand.

  6. In Eq.(12), $\chi$ looks undefined.

  7. 2nd line above Eq.(19) what does "overall sign factor" mean?

  8. At the end of 2nd paragraph of page 7, "Note that the theory is symmetric under a change of sign of the Yukawa couplings.." But the sign relative to the Wilson term on a lattice should change the physics.

  9. Eq. (22) What does this extended propagator mean?

  10. Eq (24) and the analysis follows The author should use the (numerical ) solution of cosh(m(t-T/2))/cosh(m(t+1-T/2)) = lattice data, rather than simply taking log.

  11. In page 10 2nd line from the bottom of left column, I cannot understand what "This leaves the bound state Psi" means.

---

## Round 3 · Referee Report · Anonymous (Referee 2) · 2021-1-26

Report
The authors revised the manuscript improving most of the points I suggested. However, there still remain two to be addressed.
>8.) At the end of 2nd paragraph of page 7, "Note that the theory is symmetric under a change of sign of the Yukawa couplings.." But the sign relative to the Wilson term on a lattice should change the physics.
authors> We agree that the hopping parameter has to be positive for a well-defined theory. However, our comment should only reflect that y→−y and X→−X leaves the Lagrangian in lattice (and continuum) notation unchanged.
If so, the authors should clarify X→−X transformation in addition to y→−y.
>10.) *Eq (24) and the analysis follows. The author should use the (numerical) solution of cosh(m(t-T/2))/cosh(m(t+1-T/2)) = lattice data, rather than simply taking log.*
authors>We actually did use the full lattice data without approximation in the fits as the referee suggests. We merely used the log prescription only for the plots to emphasize the onset of finite volume effects.
I know it. But the readers would be interested in the effective mass itself, rather than trivial finite volume effect on the "correlators". The (numerical) solution of cosh(m(t-T/2))/cosh(m(t+1-T/2)) should be used.
>8.) At the end of 2nd paragraph of page 7, "Note that the theory is symmetric under a change of sign of the Yukawa couplings.." But the sign relative to the Wilson term on a lattice should change the physics.
authors> We agree that the hopping parameter has to be positive for a well-defined theory. However, our comment should only reflect that y→−y and X→−X leaves the Lagrangian in lattice (and continuum) notation unchanged.
If so, the authors should clarify X→−X transformation in addition to y→−y.
>10.) *Eq (24) and the analysis follows. The author should use the (numerical) solution of cosh(m(t-T/2))/cosh(m(t+1-T/2)) = lattice data, rather than simply taking log.*
authors>We actually did use the full lattice data without approximation in the fits as the referee suggests. We merely used the log prescription only for the plots to emphasize the onset of finite volume effects.
I know it. But the readers would be interested in the effective mass itself, rather than trivial finite volume effect on the "correlators". The (numerical) solution of cosh(m(t-T/2))/cosh(m(t+1-T/2)) should be used.

Author: Vincenzo Afferrante on 2021-02-10 [id 1220]
(in reply to Report 1 on 2021-01-26)Dear editor and referee,
we are grateful for the report, and followed its suggestions:
@1: We have added a comment on the necessary transformation for the symmetry under sign-reversal of kappa in section IV.C.
@2: We added plots using this alternate definition of the effective mass in figure 1, 2, 4, and 5, and added its definition in section V, as long as a couple of additional comments in the first half of section VI.B, as this way of plotting highlights different aspects.

---

## Round 4 · Referee Report · Anonymous · 2021-2-17

Report

In the revised version, all my suggestions are reflected by the authors. I think that the paper is now good enough to be published in SciPost Physics.

---

## Round 4 · List of Changes

Updated figures 1-2-4-5, minor changes in text

---

## Editorial Decision

published